# Manufacturing of a Magnetic Composite Flexible Filament and Optimization of a 3D Printed Wideband Electromagnetic Multilayer Absorber in X-Ku Frequency Bands

**DOI:** 10.3390/ma15093320

**Published:** 2022-05-05

**Authors:** Christophe Vong, Alexis Chevalier, Azar Maalouf, Julien Ville, Jean-François Rosnarho, Vincent Laur

**Affiliations:** 1University Brest, Lab-STICC, UMR 6285, CNRS, 29200 Brest, France; christophe.vong@etudiant.univ-brest.fr (C.V.); alexis.chevalier@univ-brest.fr (A.C.); vincent.laur@univ-brest.fr (V.L.); 2University Brest, IRDL, UMR 6027, CNRS, 29200 Brest, France; julien.ville@univ-brest.fr; 3SIEPEL Cegelec Defense, 56470 La Trinité-sur-Mer, France; jf.rosnarho@siepel.com

**Keywords:** electromagnetic multilayer absorber, genetic algorithm, 3D printing, filament making, EM characterization

## Abstract

With the multiplication of electronic devices in our daily life, there is a need for tailored wideband electromagnetic (EM) absorbers that could be conformed on any type of surface-like antennas for interference attenuation or military vehicles for stealth applications. In this study, a wideband flexible flat electromagnetic absorber compatible with additive manufacturing has been studied in the X-Ku frequency bands. A multilayer structure has been optimized using a genetic algorithm (GA), adapting the restrictions of additive manufacturing and exploiting the EM properties of loaded and non-loaded filaments, of which the elaboration is described. After optimization, a bi-material multilayer absorber with a thickness of 4.1 mm has been designed to provide a reflectivity below −12 dB between 8 and 18 GHz. Finally, the designed multilayer structure was 3D-printed and measured in an anechoic chamber, achieving −11.8 dB between 7 and 18 GHz. Thus, the development of dedicated materials has demonstrated the strong potential of additive technologies for the manufacturing of thin wideband flexible EM absorbers.

## 1. Introduction

The electromagnetic (EM) wave’s problematics are becoming more crucial and ubiquitous, due to the development of electronic circuits and the rise of the Internet of Things. There is thus a prominent need for EM absorbers to reduce the impact of generated signals for various applications in different domains. For instance, the radar cross section (RCS) reductions of military aircraft and boats are vital for stealth purposes against enemy radar systems [1]; antenna devices require the mitigation of parasitic interferences from external and internal sources to operate properly [2], and high-frequency circuits require the use of microwave load to dissipate energy [3]. Besides, the performance and constraints of the absorbers may vary in terms of operating frequencies, reflectivity at various angles and polarizations, thickness, and stability in harsh environments. As such, different topologies and materials have been investigated to fabricate EM absorbers. Usually, they are designed to limit the reflection of incident waves by impedance matching, either by working on geometry-like dielectric pyramids [4], or by gradually modifying their EM properties in the case of composite absorbers [5]. Moreover, the absorption has to be increased by introducing a way to mitigate the waves inside the absorbers through different types of losses, multi-reflections, and interferences with the incident waves, such as the Dallenbach screen [6]. To increase the bandwidth, multilayer structures can be used by working on the nature and thickness of each layer, allowing for a gradual matching of the incoming waves and the introduction of various losses. These multilayer structures have the advantage of being simple to optimize by using recursive formulas proposed by Chew [7]. Many studies have used different types of algorithms to find the optimal stacking of layers resulting in various performances while keeping a thin structure, such as a genetic algorithm [8] or particle swarm optimization [9], while some [10] are looking for the optimization of the composites’ composition in a multilayer structure to achieve the same results. In order to design an ultra-wideband absorber, Li et al. [11] have managed to combine metamaterials in a multilayer structure to reach an absorption above 90% from 7.2 GHz to 35.7 GHz, and a thickness of 3.8 mm. Another paper by Pan et al. [12] also achieved a three-layer Jaumann microwave absorber by controlling the thickness of graphene oxide laminated on Kapton film, which is then placed on various dielectric substrates (foams, flexible materials), resulting in a reflection of −10 dB, covering L, S, and C bands. Finally, Kim et al. [13] have proposed a double-layer carbon metapattern using surface plasmon polaritons and coherent absorptions to achieve 90% of the absorption, from 6.3 to 30.1 GHz, with a total thickness of 4 mm. Besides, 3D printing technologies have also been investigated to realize EM absorbers using different absorbing designs. For instance, a multilayer metamaterial absorber using graphene composites has been achieved by 3D printing [14], with an absorption above 90% between 4.5 GHz and 40 GHz. Other studies from Ren et al. [15] have managed to fabricate a broadband absorber by designing dielectric resonators with a carbon-loaded acrylonitrile butadiene styrene polymer (ABS), absorbing 90% of the energy from 3.9 to 12 GHz. Without using a composites filament, research by Ghosh et al. [16] led to the design of a 3D-printed honeycomb with a PLA filament on top, which they added a resistive film to, in order to achieve a broadband absorber (at least 90%) from 5.52 to 16.96 GHz. Nevertheless, while the potential of the broadband 3D-printed EM absorber has been illustrated, the flexible flat absorber has not been investigated with this manufacturing process.

Our current goal is to obtain a flexible, flat, and thin EM absorber that could be fixed on any surface, and with the highest absorption (at least 10 dB) in the X-Ku frequency bands. The maximum total thickness has been fixed at 5 mm, which corresponds to the average thickness used in the optimization of multilayer structures [8,9]. In this article, a genetic algorithm has been developed to determine the composition of each layer of a multilayer absorber, depending on the objective and total thickness of the structure, which will then be 3D-printed by fused deposition modeling. The fabrication of a flexible loaded filament and the characterization of its EM properties from 4 GHz to 18 GHz used in the optimization are also described.

## 2. Materials and Methods

### 2.1. Materials and Filament Preparation

In this study, our aim is to demonstrate the feasibility of designing and fabricating 3D-printed wideband absorbers. The fabrication will use a fused deposition modeling method (FDM) that consists of depositing a fused polymer layer-by-layer. Flexible filaments with low loss are available, but flexible magnetic composites that could allow improving impedance matching, together with providing high absorption levels, have to be developed. Thus, composite filaments with high magnetic losses were prepared from a commercial ester-based thermoplastic polyurethane matrix selected because of its low shore hardness and its melt temperature close to 160 °C. The 3D printer used in this study is the A2V4 (from 3ntr, Oleggio, Italy), and is shown in Figure 1. This machine has a maximum printing temperature of 450 °C. All the objects were manufactured at the minimum layer thickness (i.e., 0.1 mm). Thus, the tolerance in the manufacture of the layers in the 3D printer will be of the order of ±0.1 mm. The filler used in this study was carbonyl iron particles (CIP). Studies by Abshinova et al. [17] have shown the possibility to fabricate composites with CIP and different polymers for a volume concentration from 10 to 52%. Besides, magnetic materials have better impedance matching because their permeability is higher than unity and should enhance the absorbing properties at low frequencies. While CIP could be considered to be too heavy for 3D printing, they present a higher permeability at low frequencies and broader magnetic losses than ferrites, which are slightly lighter. Moreover, this study aims to design a multilayer absorber that will not only use these magnetic composites, but also pure and porous polymers. Thus, the surface density of the absorber will be evaluated at the end of the fabrication process.

The filaments were prepared by simultaneously melt mixing all the components, polymer matrix, and CIP, using a DSM Xplore laboratory twin-screw extruder (Amsterdam, The Netherlands) at a wall temperature of 160 °C and a screw rotational speed of 150 rpm (step 1). The mixing residence time did not exceed 1 min. Samples were then freeze-fractured to obtain pellets (step 2). These pellets were incorporated in a Filabot Ex 6 filament extruder operating at a temperature of 160 °C and a fixed screw speed to control the diameter of the filament close to 2.85 mm (step 3). The same filaments melt forming process was then extrapolated on a semi-industrial and industrial scale (screw profile composed of conveying elements only, at a wall temperature of 200 °C and a screw speed of 20 rpm; coiling linear velocity of 1.6 m/min) with control of the filament diameter by a laser system. Fairly good quality for the entire filaments coil was obtained with an accuracy of 50 µm on the diameter. For the sake of clarity, the filaments elaboration process is illustrated in Table 1.

### 2.2. Characterization Techniques

#### 2.2.1. Microstructure Observation

The microstructure was observed, both before and after 3D printing, using a Hitachi S 3200 N Scanning Electron Microscope (SEM) (Naka, Japan), operating at an accelerating voltage of 15 kV. These images will be analyzed to observe the distribution of CIP in the matrix, and to note the presence, or not, of agglomerates likely to impact, in a harmful way, the electromagnetic properties of the composite. The influence of the 3D printing process on the microstructure will also be evaluated.

#### 2.2.2. Electromagnetic Characterization

EM characterization was carried out by extracting the permittivity and permeability, from the analysis of measured scattering parameters of different rectangular waveguides loaded with the samples being tested. The measurements of the scattering parameters were done with an N5245A PNA-X microwave network analyzer by Agilent Technologies (Figure 2). Then, the scattering parameters are converted using an NRW-NIST iterative method to determine both the permittivity and the permeability [18]. The samples were obtained by printing a plate with the printing parameters (thickness layer, temperature, and pattern) planned for the absorber to keep the same induced porosity of the printing parts, and to take into account the possible effects of variations on the EM properties. The different printing parameters, such as printing speed or printing temperature, as well as the dilatation or contraction effects on the object dimensions, have been adjusted for each filament with a calibration sample. The lossless filament is printed at a temperature of 230 °C, while the lossy magnetic filament is printed at 180 °C, both at the speed of 10 mm/s to ensure better control of the dimensions. The printed plate was then cut into rectangular samples to fit the dimensions of standard rectangular waveguides in C to Ku frequency bands, especially WR187, WR137, WR90, and WR62. The experimental setup is shown in Figure 2. While magnetic materials could present advantages at low frequencies thanks to their high permeability, other frequency bands have not been studied in this paper.

Besides the printing parameters used to print a fully dense object, different forms of air inclusions have been investigated. By varying the porosity, we expand the range of EM properties. This allows us to artificially increase the number of materials during the optimization, and to reach better impedance matching with the air. The size of the inclusions must be small enough, compared to the wavelength, to consider that this layer has a homogeneous material, while still being printable with a nozzle of 0.8 mm. Moreover, the air cavities must not be connected for sealing purposes. Thus, only one set of porous patterns will be investigated, to avoid printing different porous layers with various sizes or forms of the cavity on top of each other, and to guarantee the impermeability of the absorber. To not increase the printing difficulty, we only use one filament to make the porous layers. The chosen pattern is a square hole grid, with a side length of 1.8 mm and a wall thickness of 1 mm between each cell, thus adding 41% of air inclusions to the layer. The lossless filament is selected to reach a lower permittivity and to obtain better impedance matching between the air and absorber during the optimization to decrease the total reflectivity of the model. The composite filament can also be used to make porous layers, but has shown difficulties in accurately printing complex structures, and is thus not studied in this paper. A sample with the “Grid” pattern has been sliced by Ultimaker Cura (Figure 3) and printed. Then, the porous plate has been cut into rectangular samples for the EM characterization in standard waveguides.

#### 2.2.3. Measurement of the Reflectivity of the Absorber

The reflectivity measurements were done in a bistatic configuration inside an anechoic chamber, as illustrated in Figure 4, using a vector network analyzer, and wideband antennas from 4 to 18 GHz. The bistatic configuration imposes a minimum angle between the antennas of 10° in order to minimize direct coupling. Normalization was done by the measurement of a metallic plate, of which the size is the same as the sample under test. Then, the absorber was placed on the metallic plate to determine its reflection coefficient between 4 and 18 GHz. Time-gating was applied to S-parameters, in order to reduce the multi-path reflections in the anechoic chamber.

### 2.3. Structure Optimization

In order to reduce the RCS of an object, this paper describes the optimization of a multilayer structure. This absorption method has the advantage of combining non-loaded material for impedance matching with the incident EM wave, and materials with dielectric/magnetic losses to dissipate the energy. EM characterization will allow us to extract the permittivity *ε* and the permeability *μ* of each material:(1)$$\varepsilon =\varepsilon^{\prime }-i\varepsilon^{\prime \prime }=\varepsilon_{0}\left(\varepsilon_{r}^{\prime }-i\varepsilon_{r}^{\prime \prime }\right)$$
(2)$$\mu =\mu^{\prime }-i\mu^{\prime \prime }=\mu_{0}\left(\mu_{r}^{\prime }-i\mu_{r}^{\prime \prime }\right)$$
with $\varepsilon_{r}^{\prime }$, $\varepsilon_{r}^{\prime \prime }$, $\mu_{r}^{\prime }$ and $\mu_{r}^{\prime \prime }$ being the real and imaginary part of the permittivity and permeability, and $\varepsilon_{0}$, $\mu_{0}$ those of the vacuum.

With the data from the EM characterization, a genetic algorithm (GA) has been used to determine the nature of the different layers in our multilayer EM absorber. However, the configuration of the machine for printing flexible materials only allows the use of a maximum of 2 nozzles, and thus we have decided to start the optimization by using 2 materials: the lossless filament (with or without air inclusions) and the lossy magnetic filament. By considering the resolution and by fixing the thickness of the objective structure, it is possible to slice the system into a fixed number of layers (for instance, a total thickness of 10 mm with a maximum resolution of 0.1 mm will result in 100 sublayers). Different thicknesses are tested to find the thinnest structure to fulfill the goal. We set the following rules for the GA optimization of the multilayer structure. The material library will be composed of the EM properties of:The lossless filament when fully dense;The magnetic loaded filament when fully dense;The lossless filament with the “Grid” porous pattern.

Moreover, the first and last layers are constrained to be made of a material without porosity to ensure the impermeability of the absorber. Since better impedance matching leads to better microwave absorption, the first layer is fixed to be made by the non-loaded filament, while the last layer is fixed to be made by the magnetic loaded filament. As for the rest of the layers, the GA algorithm creates a vector population determining different combinations of layers, based on the Z resolution and the thickness chosen, where each individual contains the nature of each layer as “genes”. The genes of each vector will be adjusted after each iteration to find the optimal structure. The Z resolution of a 3D printer refers to the minimum layer thickness that can be printed. Each vector will thus contain a number of genes equaling the total thickness of the absorber without the first and the last layers, divided by the resolution of the 3D printer (0.1 mm in this paper). The size of the population was fixed at 100. The multilayer is backed by a perfect electric conductor (PEC), as shown in Figure 5a. The fitness function, also called the figure of merit, allowing the sorting of the different possibilities, is the difference between the area of the reflectivity calculated with each vector, and that of the goal reflectivity, which is first set at −10 dB in the X-Ku frequency bands at 10°. To calculate the global reflectivity coefficient ${\tilde{R}}_{0,1}$, we need the reflection coefficient at each i interface $R_{i}$ of the multilayer using these equations with the EM properties of each layer,$\;\mu_{i}$ for the permeability and $\varepsilon_{i}$ the permittivity at the layer *i*:(3)$$for\;TE\;polarization:\;R_{i,i+1}=\frac{\mu_{i+1}k_{i}-\mu_{i}k_{i+1}}{\mu_{i+1}k_{i}+\mu_{i}k_{i+1}},$$
(4)$$for\;TM\;polarization:\;R_{i,i+1}=\frac{\varepsilon_{i+1}k_{i}-\varepsilon_{i}k_{i+1}}{\varepsilon_{i+1}k_{i}+\varepsilon_{i}k_{i+1}}$$
where $k_{i}=\omega \sqrt{\mu_{i}\varepsilon_{i}-\mu_{0}\varepsilon_{0}sin^{2}\left(\theta \right)}$, ω being the angular frequency, θ the angle of incidence, $\mu_{0}$ the vacuum permeability and $\varepsilon_{0}$ the vacuum permittivity.

Then, we can use the recursive formula of Chew using these coefficients and the thickness $d_{i}$ of each layer to find the generalized reflection coefficient between each layer and of the entire structure:(5)$${\tilde{R}}_{i,i+1}=\frac{R_{i,i+1}+{\tilde{R}}_{i+1,i+2}e^{-j2k_{i+1,z}d_{i+1}}}{1+R_{i,i+1}{\tilde{R}}_{i+1,i+2}e^{-j2k_{i+1,z}d_{i+1}}}$$
where ${\tilde{R}}_{n,n+1}=-1$ for TE (transverse electric) polarization, and ${\tilde{R}}_{n,n+1}=+1$ for TM (transverse magnetic) polarization at the interface between the last layer of material and the PEC. As mentioned by Dib et al. [19], because Equation (4) represents the reflection coefficient for the magnetic field for TM instead of the electric field for TE, ${\tilde{R}}_{n,n+1}$ has to be set to à +1 to reach the same magnitude with both equations at normal incidence.

After the selection, the best vectors are kept and combined to create new combinations, creating another population with more optimal solutions. The algorithm proceeds until it reaches the objective or after meeting stopping criteria, such as converging toward a local minimum or reaching a fixed generation, fulfilling the objective criteria (Figure 5b). As for the other parameters of the GA algorithm, every function is set up with default options in the Matlab optimization toolbox [20]. For instance, the stochastic uniform option is used for the selection function, the Gaussian mutation is set at a rate of 0.01 between each generation, a scattered crossover option is chosen to create new combinations of layers, and the maximum generation is limited by 100× number of genes.

## 3. Results

### 3.1. Microstructure Observation

The CIP has a spherical aspect with a characteristic size of a few microns, as shown on the scanning electron micrograph in Figure 6. In particular, the average diameter, estimated from about 300 individual measurements using Sigman Scan^®^ PRO 5 images analysis software, is close to 1.8 µm, with a standard deviation of 0.8 µm, as illustrated on the diameter distribution curve in Figure 7. This value is close to the one given by the supplier. In the paper by Abshinova et al. [17], different concentrations of CIP have been tested from 10% to 52% vol. However, the composite filament must show a permanent flow at the exit of the extruder to ensure the potential ability to process the samples. If the flow is not permanent, the 3D printing deposition cannot be controlled, and the printed object will suffer from dimensional inaccuracies. Thus, we have to find a compromise between having the highest concentration to ensure the best absorbing performances, while being printable without flowing issues. After testing different composites, a high filling ratio of carbonyl iron particles (75% wt or 30% vol) was chosen to obtain a flexible elastomer/CIP composite filament with high magnetic losses. A higher concentration could be tested in the future.

As shown in Figure 8a,b, most CIP are well dispersed within the matrix without a significant difference between the microstructure of the 3D-printed sample (Figure 8b), and that of the filament used for 3D printing (Figure 8a). Besides, the absence of aggregates is attributed to the high frequency of collisions between CIP during the extrusion process estimated from the Smoluchowski equation [21,22]:(6)$$C=\frac{8}{\pi }\dot{\gamma }\varphi ,$$
where *C* is the frequency of collisions, $\dot{\gamma }$ the shear rate, and φ the volume fraction of CIP. If we consider that the average shear rate is equal to 2200 s^−1^, estimated from the technical data, during mixing, then *C*~200 collisions per second at φ = 10% vol, and *C* reaches 1700 collisions per second at φ = 75% wt. The frequency of collisions is the number of collisions between particles per unit time. A high number of collisions allow one to disrupt the formation of aggregates.

### 3.2. Microwave Characterization

The measured EM properties for the two filaments are presented in Figure 9. According to the EM characterization, the filament with carbonyl iron filler shows a high magnetic loss tangent of around 0.5–1 (Figure 9d), and can thus act as a microwave absorber. The real part of the permeability decreases from 2.4 to 1.1 in the 4–18 GHz frequency band. The real permittivity (Figure 9c) is around 10 and is very stable among the frequency band under study. The lossless filament has a permittivity between 2.5 and 3 that slightly decreases with increasing frequency (Figure 9a). As illustrated in Figure 9e, the air inclusion addition effectively reduces the permittivity of the porous structure compared to the filament when fully printed, from around 2.8 to 1.8 for the real part of permittivity. These values of EM properties were used for the optimization of the multilayer microwave absorber. To evaluate the repeatability of the printing process, the tolerance was calculated by characterizing different samples for each material. In the case of the non-loaded filament, the tolerances of the permittivity are ±0.3 for the real part and ±0.02 for the imaginary part. For the magnetic-loaded filament, the tolerances of the permittivity are ±0.8 for the real part and ±0.3 for the imaginary part, and those of the permeability are ±0.08 for the real part and ±0.08 for the imaginary part.

### 3.3. Structure Optimization

Several levels of absorption have been tested in the X-Ku band for multilayer structures with a maximum thickness of 5 mm. Figure 10 presents the simulated results of a multilayer structure made from the lossless material and the material with magnetic losses, with a goal of −12 dB in the X-Ku frequency bands at 10° with and without a fixed number of alternating layers. This angle of incidence was chosen because it corresponds to the closest angle to the normal incidence of our bistatic measurement setup. Optimizations of a higher level of absorption (more than 13 dB) in these bands require a higher thickness and will not be described. Using the optimization method as described in Section 2.3, a multilayer structure with 15 alternating layers was found, which could be difficult to print, especially because of the fact that porous layers have to be well defined, and most of the layer thicknesses are thin. Thus, another optimization has been carried out by fixing the number of layers at 3 to ease the printing process, and allow us to obtain a first experimental demonstration of a multilayer absorber before printing more complex structures. The nature of each layer for these structures, as well as each thickness, is described in Table 2 and Table 3.

The thinnest structure to fulfill the goal of −12 dB was found to be 4 mm. However, the optimization method leads to a multilayer with 15 different layers that might be difficult to print accurately. Moreover, this optimized multilayer absorber has alternating very thin layers of porous and non-porous non-loaded materials that can be replaced by thicker alternating layers to achieve impedance matching. Thus, by fixing the number of layers at 3, a similar result can be obtained with a thickness of 4.1 mm. The structure being quite simple, it seems that a wideband thin absorber over the X and Ku frequency bands might be feasible to print based on the simulation. In this case, the 3D printing process will allow for a single-step process to manufacture a microwave absorber, as well as controlling the porosity of one of the layers to impact its EM properties.

### 3.4. 3D Printing

Based on the optimization of the multilayer structure, a square model with a size of 100 mm × 100 mm of the EM absorber was sliced by Ultimaker Cura and printed. Because of the flexibility of the composite filament, the printing speed has been limited to 10 mm/s, thus reducing the time to print the multilayer to under 24 h. Small extensions on the sides have been made to estimate the thickness of each layer with a caliper; they are then removed before measuring the reflectivity inside an anechoic chamber. After measurements of the absorption, the sample is sliced to measure the thickness of each layer with a microscope for better accuracy. Models and pictures of the printed objects are presented in Figure 11. The surface density was 5.9 kg m^−2^, which is lighter compared to the Eccosorb SLJ by Laird Technologies (8 kg m^−2^), a commercial flexible wideband absorber with a thickness of 6.7 mm which does not use magnetic materials. Since the magnetic layer only represents 30% of the total thickness of the optimized absorber, the surface density of the absorber is not considerably heavier than other commercial absorbers.

The measurement in the anechoic chamber of the absorber was made in a bistatic configuration, at an angle of incidence of 10°, for both linear polarizations TE and TM, and can be seen in Figure 12, as well as the simulations with the same configuration. The measurement was made on sample with the measured values for each of the layers given in Table 4. Based on the measurement, the absorber follows the expected behavior and reaches −11.8 dB between 7 GHz and 18 GHz (or −10 dB between 6.5 GHz and 18 GHz).

## 4. Discussion

The result demonstrates the potential of additive manufacturing for the development of flexible broadband absorbers. However, we observe a shift towards lower frequencies of around 2 GHz compared to the optimization, even though the structure seems to be thinner than the optimization. It is likely that the dimension of each layer does not exactly match what has been simulated, and a study of the sensitivity is required to assess this hypothesis. Concerning the bottom layer, the slight thickness differences are surely due to the offset between the nozzle and the 3D printer bed, and the other slight variations might be begotten by the porosity of the layer and the flexibility of the structure. Moreover, the concentration of the air inclusion seems difficult to evaluate, but we will have to consider possible variations of the EM properties of this layer.

As for the study of the sensitivity, the error impact caused by the printing resolution of the structure has been studied. We have also considered the variation of the diameter of the filament (50 µm), which can change the material flow through the nozzles and impact the thickness of each layer. Finally, we have also evaluated the possibility of a variation of the filler concentration or uncertainty of the EM properties, which can impact the performances of the structures. To evaluate the influence of such parameters, several parametric studies have been performed on the optimized structure using ANSYS HFSS software.

First, we have looked at the impact of a dimensional error on each layer thickness. As such, we choose to modify the thickness of each layer by more or less 0.1 mm, which is equal to the resolution of the machine, and thus will be considered as the maximum error that we will first accept. Based on the simulations (Figure 13a–c), it seems that the main parameter affecting the sensibility of the system is the thickness of the last layer (magnetic loaded material). If the bottom layer is thicker, the bandwidth will be larger, but the maximum reflectivity will degrade up to −11 dB in bandwidth. Otherwise, we should observe the opposite and expect a smaller bandwidth, but a more absorbing structure. However, because the overall structure, as well as the thickness of the bottom layer, are thinner than the simulation, this should result in a frequency shift toward higher frequencies. The deviation should be explained by the other factors.

Then, we looked at the influence of the filler concentration variation in the material. Because making a new filament for every concentration would be too cumbersome, we decided to modify the EM properties (permittivity and permeability at the same time) of the magnetic filament by a fixed percentage (more or less 10%, EM properties shown in Figure 13e,f) to evaluate the impact of such variation on the reflectivity. According to the results in Figure 13d, EM properties’ variations of the magnetic loaded filament have a non-negligible impact, with a possible frequency shift of 1 GHz, with variations of the EM properties by only 10%. If the filler concentration is higher, then we should expect a shift of the absorption bandwidth toward lower frequencies and a reduction of the absorption in the X and Ku frequency bands, and we should see the opposite in the other case.

Other than this, we consider the variation of the porous layer EM properties. Indeed, the EM characterization has been done without any material printed on top of the grid, which could lead to different results compared to the printing, where the top layer might fill the air inclusion. As such, the EM properties of the intermediate layer may be between those of the lossless material when fully printed, and those with air inclusions. To evaluate such an issue, we consider the worst case being the porous lossless layer replaced by the fully printed lossless layer. By looking at Figure 13g, it seems that we are expecting a frequency shift toward the lower frequencies, as well as a degradation of the absorption if the porous layer is not printed correctly.

Finally, by studying the influence of an air gap under the structure due to internal stresses or the roughness after printing, we can see in Figure 13h that this issue leads to a frequency shift and the degradation of the global absorption. As such, it will be important to check during the measurements that the structure is as flat as possible, without using any tape or glue to fix the structure, since they will increase the gap.

According to the above study, a combination of a higher magnetic loss from the loaded material, a possible reduction of the air inclusion in the porous layer, and an addition of a small air gap due to the roughness of the structure might explain the frequency shift compared with the optimization. Thus, it is necessary to search for a way to control the dimension of each layer thickness and the air inclusion of the porous layer, especially with flexible filament.

## 5. Conclusions

In summary, the manufacturing of a 3D-printed multilayer broadband EM absorber has been described from the creation of the filaments enabling magnetic EM losses to the optimization with a genetic algorithm, taking into account the possibilities offered by FDM 3D printing, but also its limitations. A 4.1-mm-thick absorber with a bandwidth between 7 to 18 GHz in which return losses are lower than −11.8 dB has been optimized and measured in an anechoic chamber, thus demonstrating the potential of this dedicated filament and of the 3D printing technique to fabricate an efficient flexible EM absorber. Future studies will consist of improving the current 3D printing parameters in such a way that it can use three different flexible materials instead of two today, as well as enhancing the accuracy of the thickness of the different layers, which will be crucial with more complex structures. However, the preferred route initially will be the use of materials with higher iron filling content, which will necessarily require further studies on the thermomechanical behavior of the composite thus obtained.

## Figures and Tables

**Figure 1 materials-15-03320-f001:**
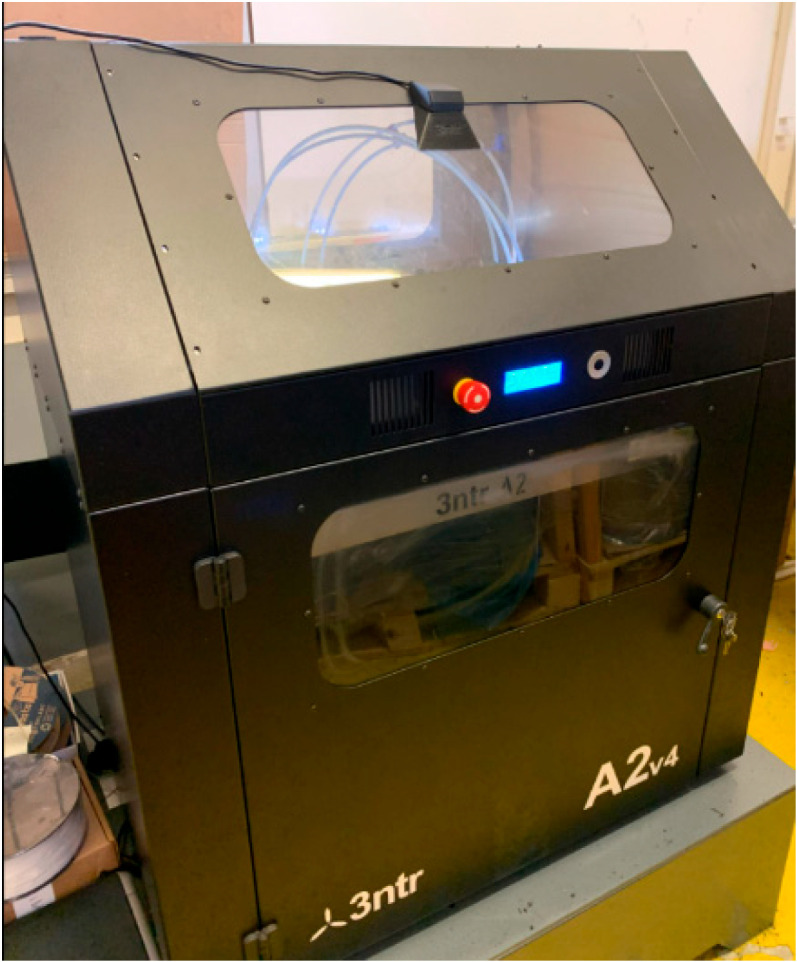
Picture of the 3D printer A2V4 from 3ntr.

**Figure 2 materials-15-03320-f002:**
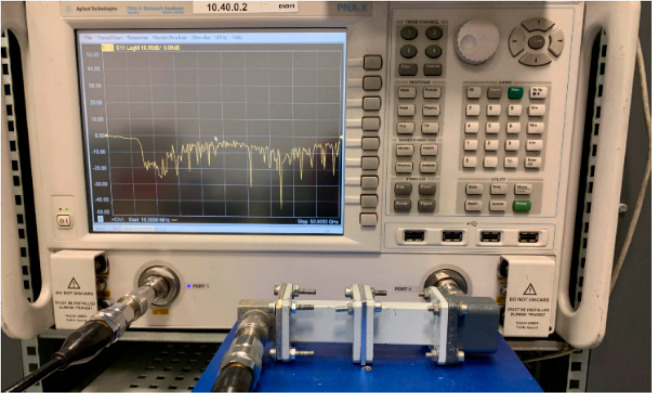
Experimental setup of the waveguide characterization of the printed samples.

**Figure 3 materials-15-03320-f003:**
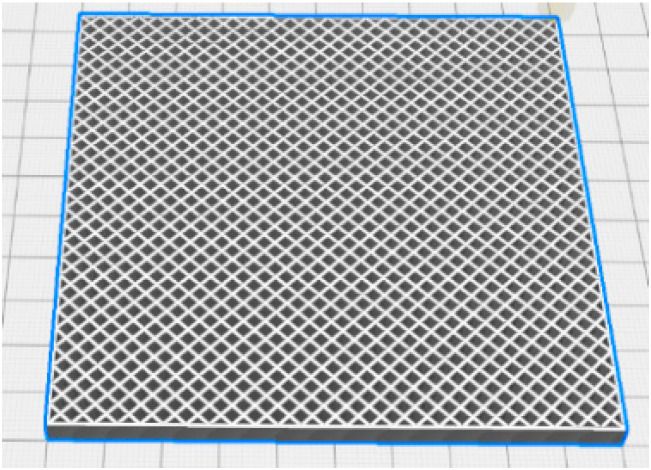
Image of a sample with the “Grid” pattern sliced in Ultimaker Cura, with an “infill line pattern” at 2.8 mm (1.8 mm for the size of the holes and 1 mm for the wall).

**Figure 4 materials-15-03320-f004:**
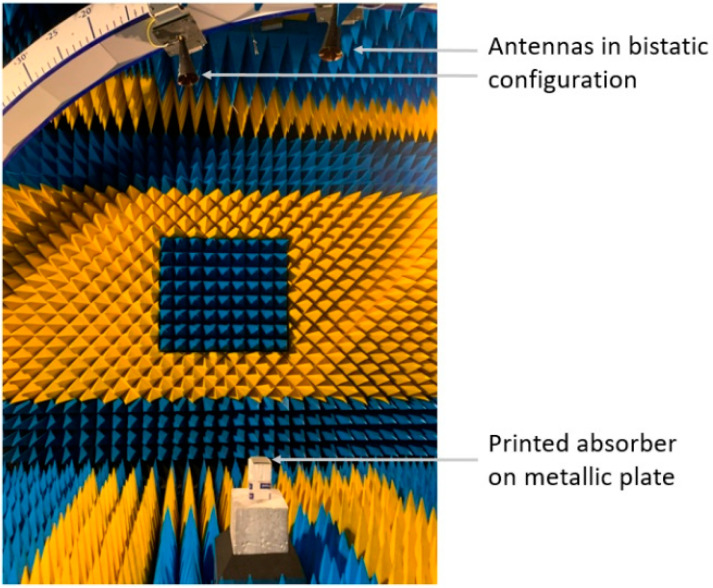
Photography of the setup process of the reflectivity measurement of the absorber in the anechoic chamber.

**Figure 5 materials-15-03320-f005:**
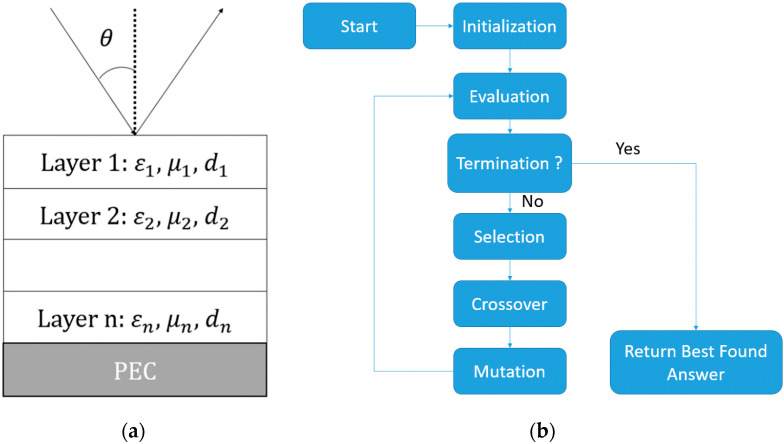
(**a**) Multilayer absorber structure; (**b**) flowchart of the GA optimization.

**Figure 6 materials-15-03320-f006:**
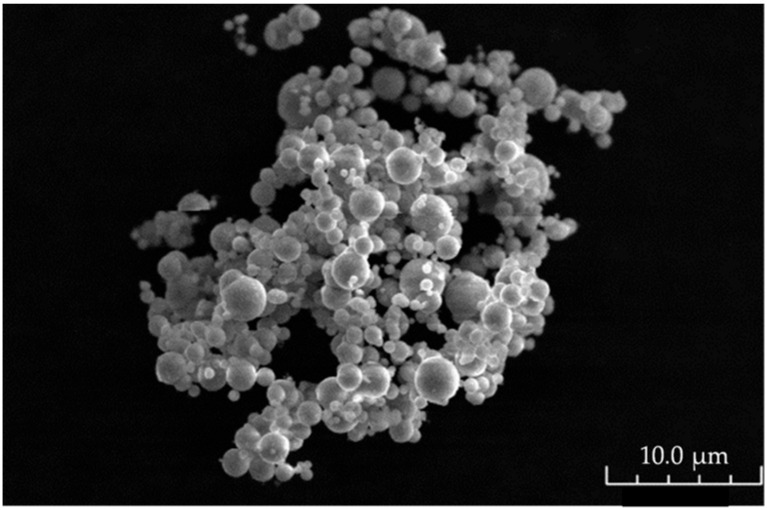
Scanning electron microscopy micrograph of CIP.

**Figure 7 materials-15-03320-f007:**
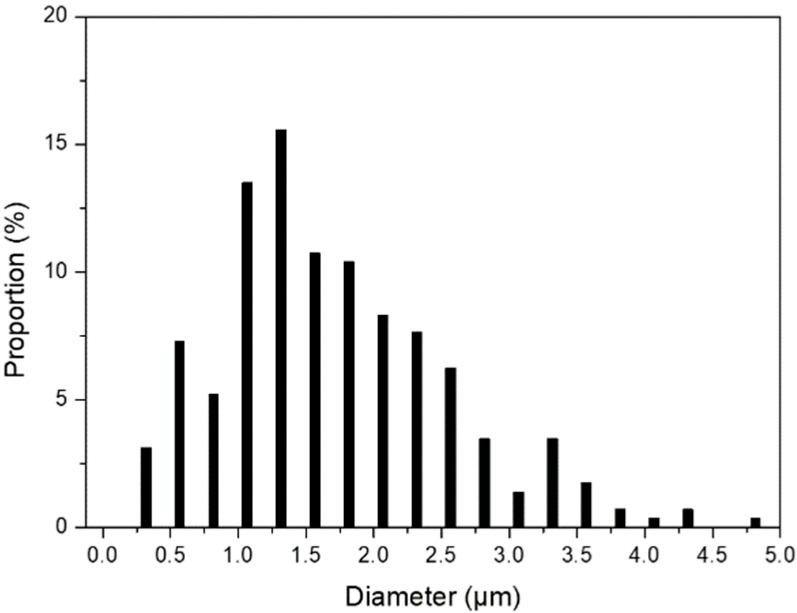
Diameter distribution curve for CIP.

**Figure 8 materials-15-03320-f008:**
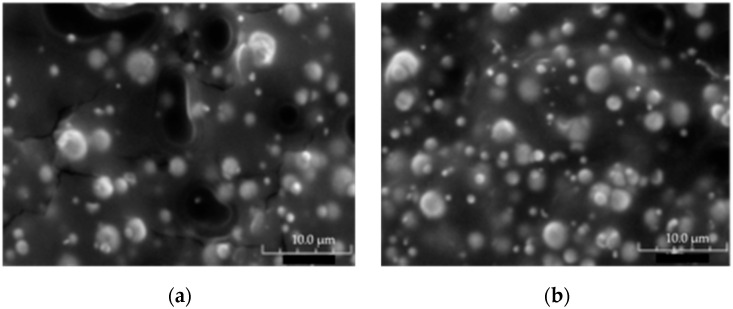
Scanning electron microscopy images of (**a**) the magnetic filament, (**b**) printed sample with the magnetic filament.

**Figure 9 materials-15-03320-f009:**
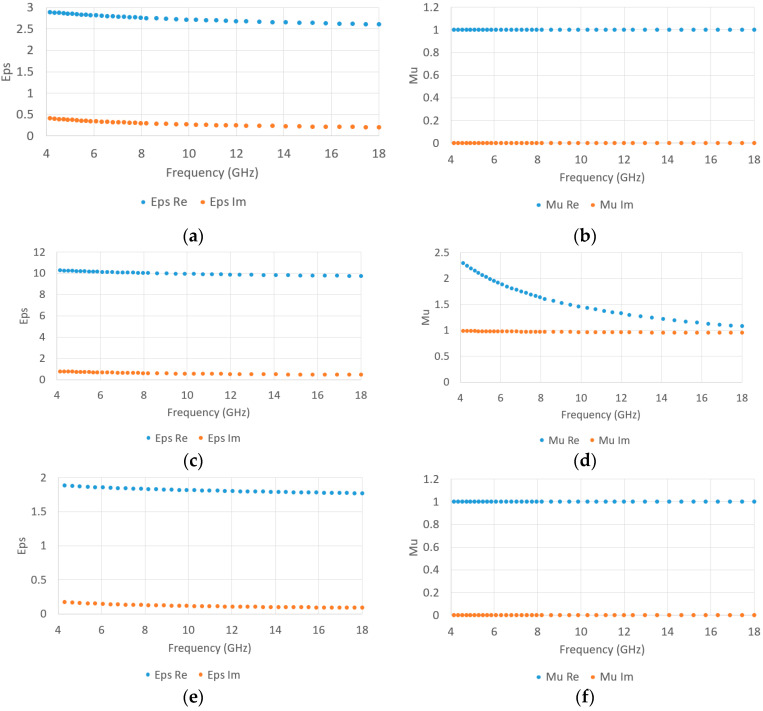
Characterization of the EM properties, with Eps Re = real permittivity, Eps Im = imaginary permittivity, Mu Re = real permeability, Mu Im = imaginary permeability: (**a**) the permittivity of the non-loaded filament, (**b**) the permeability of the non-loaded filament, (**c**) the permittivity of the magnetic loaded filament, (**d**) the permeability of the magnetic loaded filament, (**e**) the permittivity of the printed sample with a “Grid” pattern, (**f**) the permeability of the printed sample with a “Grid” pattern.

**Figure 10 materials-15-03320-f010:**
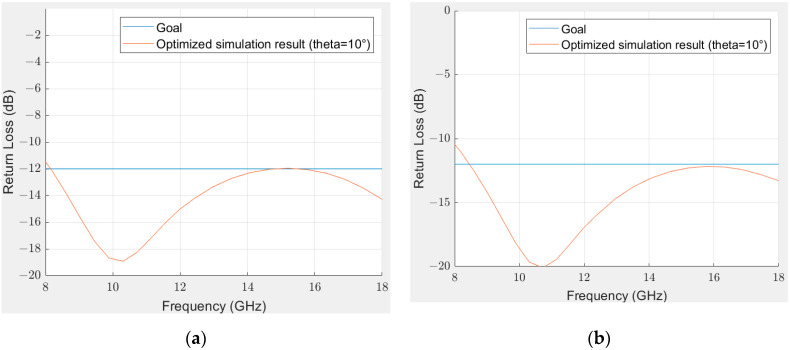
Reflectivity of the optimized structure in Matlab with a goal of −12 dB at 10° (blue = goal, red = RL) of the optimized structure at 10°, (**a**) without restrictions on the number of different layers, (**b**) number of layers fixed at 3.

**Figure 11 materials-15-03320-f011:**
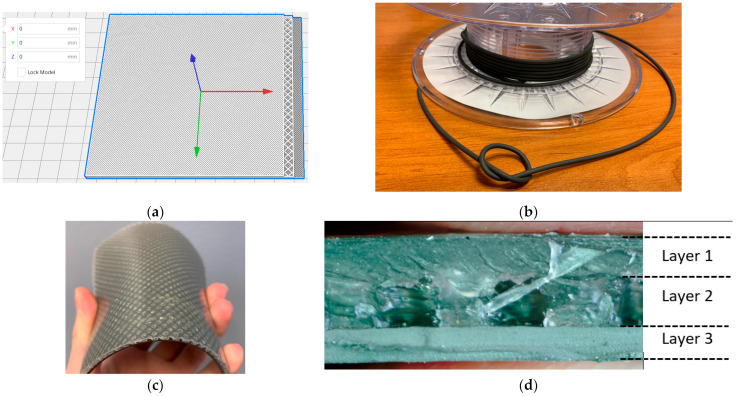
(**a**) Model of the multilayer structure sliced in Ultimaker Cura (white = non-loaded filament, grey = magnetic loaded filament, red axis = X, green axis = Y, blue axis = Z), (**b**) picture of the flexible composite filament, (**c**) picture of multilayer EM absorber after printing and without extensions, (**d**) picture of the multilayer on the side.

**Figure 12 materials-15-03320-f012:**
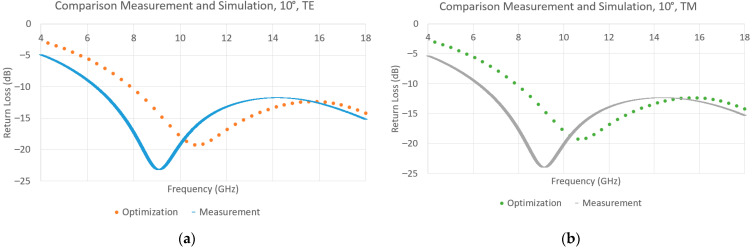
Result of the measurement of the multilayer between 4 GHz and 18 GHz and comparison with the simulation of the optimized structure at 10°, (**a**) TE, (**b**) TM.

**Figure 13 materials-15-03320-f013:**
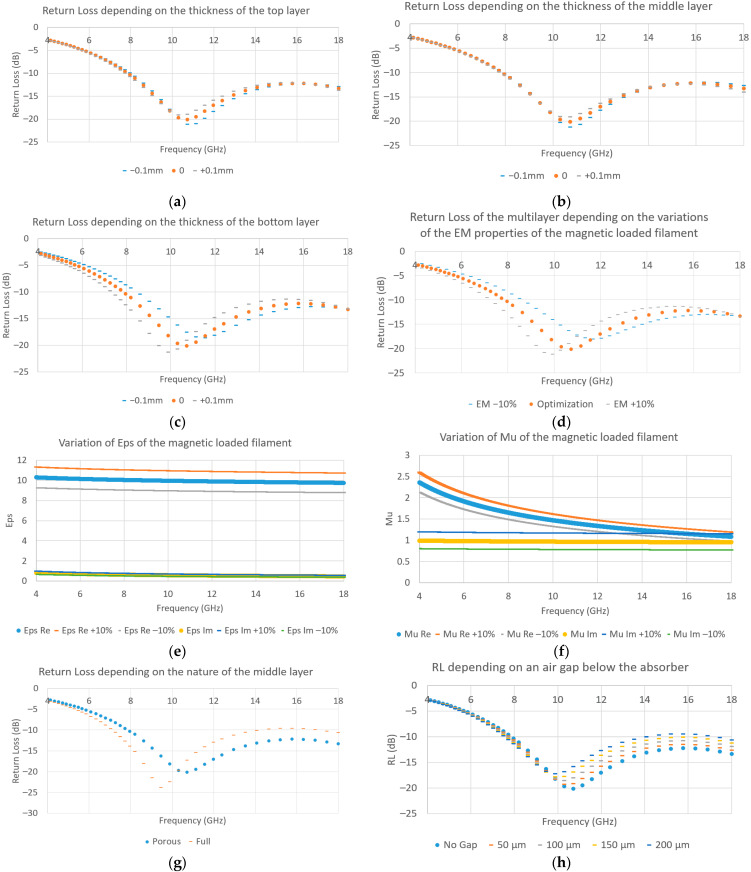
Parametric study of the thickness on the return loss (RL) at 10° of (**a**) the top layer, (**b**) the middle layer, (**c**) the bottom layer, (**d**) comparisons of simulations using different variations of EM properties of the magnetic-loaded material, variations of the EM properties of the magnetic loaded filament (**e**) of the permittivity Eps, (**f**) of the permeability Mu, (**g**) comparison of the reflectivity by modifying the nature of the middle layer, porous or full, (**h**) parametric study of an air gap under the absorber, 50 µm to 200 µm.

**Table 1 materials-15-03320-t001:** Scheme of global filaments elaboration process.

	I Simultaneous Melt Mixing Using a Twin-Screw Extruder	II Pellets Processing	III Filament Processing	
Laboratory scale	Twin-screw extrusion T = 160 °C, N = 150 rpm	(1) Nitrogen cooling (2) Pelletizing	(1) Extrusion (conveying) T = 200 °C, N = 20 rpm (2) Coiling v = 1.6 m/min	(3) Air cooling φ = 2.85 mm ± 500 µm
Semi-industrial scale				(3) Water cooling φ = 2.85 mm ± 200 µm
Industrial-scale				(3) Water cooling φ = 2.85 mm ± 50 µm

**Table 2 materials-15-03320-t002:** Composition of the optimized solution without restrictions on the number of layers.

N° Layer	Composition	Thickness (mm)
1	Non-loaded	0.3
2	Magnetic loaded	0.1
3	Non-loaded	0.2
4	Non-loaded with porosity	0.2
5	Non-loaded	0.1
6	Non-loaded with porosity	0.7
7	Non-loaded	0.2
8	Non-loaded with porosity	0.2
9	Non-loaded	0.1
10	Non-loaded with porosity	0.1
11	Non-loaded	0.1
12	Non-loaded with porosity	0.1
13	Non-loaded	0.2
14	Non-loaded with porosity	0.2
15	Magnetic loaded	1.2

**Table 3 materials-15-03320-t003:** Composition of the optimized solution.

N° Layer	Composition	Thickness (mm)
1	Non-loaded	1.3
2	Non-loaded with porosity	1.6
3	Magnetic loaded	1.2

**Table 4 materials-15-03320-t004:** Composition of the thickness measured for each printed layer.

N° Layer	Optimized Thickness (mm)	Measured Thickness (mm)
1	1.3	1.25
2	1.6	1.81
3	1.2	1.05

## Data Availability

Not applicable.

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
