# Peer review of "Manufacturing of a Magnetic Composite Flexible Filament and Optimization of a 3D Printed Wideband Electromagnetic Multilayer Absorber in X-Ku Frequency Bands"

_materials, 2022, doi:10.3390/ma15093320_

Round 1

Reviewer 1 Report

File attached

Reviewer 2 Report

This paper implements a flexible absorber material in the X-Ku frequency bands by designing and fabricating magnetic materials using a multilayer structure and 3D printing techniques. The work done is interesting and meets the standards of the journal materials, but needs major revision before it can be published in the journal. 

Minor comments:

Include some more reference in the introduction to 3D printing technology in the manufacture of absorbent materials. 

In Figure 7 instead of Fréq it should be Freq or Frequency.

Figure 8 Frequency instead of Frecuence

Determine each material in Figure 8.

Specific comments:

Why is the target 5 mm layer, justify in the document? Line 50

Could you include a picture of the setup process of the electromagnetic characterisation of the materials?

What is the tolerance in the permittivity and permeability characteristics of the filaments analysed, in terms of repeatability in the printing and manufacturing process on the 3D printer?

Which 3D printer has been used in the manufacture of the absorber materials? could you please include a picture of the printer?

Could you include a picture of the setup process of the reflectivity measurement of the absorber part in the anechoic chamber? 

What is the tolerance in the manufacture of the layers in the 3D printer?

What equipment is used to measure the reflectivity of the absorber?

Reviewer 3 Report

The paper deals with the 3D printed multilayer absorber, which is of general interest for researchers working on absorbers. The paper is largely fine, but the following major issues need to be addressed:

(1)The multilayer structure by 3D printing presented in the paper indeed shows good MA performance, but it is more desirable for the authors to design more complicated structure which can highlight the advantage of the 3D printing techniques over conventional ones. For the current structure, what are the advantages of using 3D printing?

(2)CIP is indeed good absorbent, but it is too heavy, and I do not think CIP favors 3D printing, since too much of it will make the printing not possible and also the absorber very heavy. The authors need to comment on the rationale for their choice of CIPs. It is also necessary to give the density or specific areal density of the absorber.

(3)MA at low frequency remains a challenge, it is hoped that magnetic absorbents with large mu and controllable domain structure could solve the issue. Can the authors provide an optimized design at, say, 1-6GHz for good absorption?

Reviewer 4 Report

In this manuscript, a design and an additive manufacturing method for wideband electromagnetic multilayer absorber are proposed. Even though the proposed bi-material multilayer absorber has novelties from the perspective of fabrication methodology as well as flexibility of the structure, there are many omitted information about the design strategy as well as data to support the absorbing performance for the bended structure. In my opinion, the authors have to improve the completeness of the manuscript significantly.

  1. In my opinion, the authors should revise the title to represent the novelties of the manuscript clearly.
  2. I recommend the authors to supply recently published papers related with multilayer electromagnetic absorbers such as J. Appl. Phys. 122, 055104, (2017), Carbon 160, pp. 307–316, (2020), and Sci. Rep. 11, 23045, (2021) as well as to appeal the novelties of the proposed absorber clearly comparing with the previous papers. In my opinion, it may be needed to extend the frequency range of the simulation above 18 GHz to appeal the merit of the proposed absorber from the perspective of the -10 dB bandwidth.
  3. I cannot understand clearly the reason why the incident angle θ should be confined to 10°. If the authors measured the return loss of the fabricated sample in the anechoic chamber, there is no restriction to increase or decrease the incident angle. Please provide the information about the measurement set up detailedly. To appeal the flexibility as one of the novelties of the proposed absorber, the authors have to supply at least the simulation results of the return loss for various incident angles with the TE and TM polarizations and/or of the monostatic radar cross section of the bended structure. Please find a recommended paper Sci. Rep. 9, 12334, (2019).
  4. Please provide the derivations or references to support the equations (1), (2), and (3). I do not believe the reflection coefficient Rn,n+1 is different for the TE and TM polarizations at the interface between the last layer and the PEC . It should be always -1.
  5. Please provide detailed methodology or references utilized to model the constitutive parameters epsilon (ε) and mu (μ) of each layer in Fig. 7. How did the authors determine the ratio of carbonyl iron particle (CIP) 75% as the optimum one?
  6. It is needed to supply more specific parameters utilized for the genetic algorithm (GA) such as the number of generated gene set, iteration number, number of operations for crossover and mutation, figure of merit used for the competition process, etc.
  7. It is needed to provide more detail explanation about the simulation setting in Fig. 8(b). A reader cannot understand what the authors describe about it. In my opinion, there exist unnecessary vertical lines for “goal” in Fig. 8(a) at 8 and 18 GHz. It is recommended to revise the legend in Fig. 8(a) from “10°” to “Optimized simulation result (θ=10°)”. The name of x axis in Fig. 8(a) has to be revised from “Frequence (GHz)” to “Frequency (GHz)”.
  8. I wonder whether the authors used any condition to confine the number of layer to three or it is automatic result found by GA.
  9. Minor comments: 
    • Please provide the full names of EM where it is firstly mentioned in Introduction, SEM in Figs. 4 and 6, Eps in Fig. 7, and RL in Fig. 11. 
    • What does the “frequency of collisions” in line 183 mean? 
    • Please revise the explanation about ω from “pulsation of the wave” to “angular frequency”.
    • Please provide the x, y, and z axes in Fig. 9(a) clearly. 

Round 2

Reviewer 2 Report

The authors have responded to my suggestions and comments and the article is ready for publication in the journal, I suggest correcting some spelling mistakes in the text.

Author Response

Dear Reviewer,

Thank you for taking the time to review our article one more time. We have taken your suggestion into account and have tried correcting spelling mistakes in the text.

Best regards,

Reviewer 3 Report

The authors have adequately addressed my previous comments, i therefore recommend its publication as it is.

Author Response

Dear Reviewer,

Thank you for taking the time to review our article one more time.

Best regards,

Reviewer 4 Report

I would like to thank the authors for their efforts on manuscript revision. The manuscript was improved in many places and the criticisms raised from the previous version were almost solved by the authors' response. However, the manuscript has to be improved further by following some comments listed below.

  1. Even though the authors have supplied information about genetic algorithm (GA), I found that there remain incomplete sentences in the manuscript.
  • In line 211, what does the “Z resolution” mean?
  • In line 212, what do the “X genes” and “X equaling” mean?
  • Please add the reference for the “Matlab optimization toolbox” in line 237.
  • It is found that the sentence in line 239 is not completed.
  1. Even though the authors have supplied the definition of the constitutive parameters of the material from lines 183 to 189, it is not clear how they extract the results shown in Fig. 10 from the waveguide measurements. Please provide the methodology in detail or supply some references for the procedure.
  2. In my opinion, Table 1 should be deleted because it is unnecessary. The bandwidths and the sizes of the standard waveguides are well known. By just mentioning the name of the waveguides used for the measurements in the manuscript, the reader can understand the experimental conditions.
  3. Minor comments:
  • In Fig. 5, please remove the purple under lines shown at “bistatic”, “printed”, and “metallic”.
  • In line 93, please remove the unnecessary grey shade at “±”.
